REGISTERED REPORT PROTOCOL

# Developmental fronto-parietal shift of brain activation during mental arithmetic across the lifespan: A registered report protocol

Christina Artemenko [1,2] *

**1** Department of Psychology, University of Tuebingen, Tuebingen, Germany, **2** LEAD Graduate School & Research Network, University of Tuebingen, Tuebingen, Germany

* christina.artemenko@uni-tuebingen.de

## Abstract

Arithmetic processing is represented in a fronto-parietal network of the brain. However, activation within this network undergoes a shift from domain-general cognitive processing in the frontal cortex towards domain-specific magnitude processing in the parietal cortex. This is at least what is known about development from findings in children and young adults. In this registered report, we set out to replicate the fronto-parietal activation shift for arithmetic processing and explore for the first time how neural development of arithmetic continues during aging. This study focuses on the behavioral and neural correlates of arithmetic and arithmetic complexity across the lifespan, i.e., childhood, where arithmetic is first learned, young adulthood, when arithmetic skills are already established, and old age, when there is lifelong arithmetic experience. Therefore, brain activation during mental arithmetic will be measured in children, young adults, and the elderly using functional near-infrared spectroscopy (fNIRS). Arithmetic complexity will be manipulated by the carry and borrow operations in two-digit addition and subtraction. The findings of this study will inform educational practice, since the carry and borrow operations are considered as obstacles in math achievement, and serve as a basis for developing interventions in the elderly, since arithmetic skills are important for an independent daily life.

## Introduction

Arithmetic skills are acquired in school and are later important for everyday life. Hence, it is essential to better understand the underlying mechanisms of these skills not only in children, who just learnt these skills, and in adults, who have already established their arithmetic skills, but also in the elderly, because deficits in these skills have a detrimental impact on their independent life. Therefore, the current study sets out to investigate the behavioral and neural correlates of arithmetic across the lifespan.

Arithmetic is represented in a fronto-parietal network in the brain [1, for meta-analyses in children and adults see 2, for a review in children see 3, for a model and its extensions in adults see 4, 5], but the extent of frontal and parietal activation in this network changes during

**Data Availability Statement:** This is a Registered Report Protocol without pilot data; data will be shared after data collection.

**Funding:** This research project was funded by the Institutional Strategy of the University of Tuebingen (Deutsche Forschungsgemeinschaft, ZUK 63). This research project was further supported by the European Social Fund and the Ministry of Science, Research and the Arts Baden-Wuerttemberg, by the LEAD Graduate School & Research Network (GSC1028, funded within the Excellence Initiative of the German federal and state governments), and by the Tuebingen Postdoc Academy for Research on Education (PACE) at the Hector Research Institute of Education Sciences and Psychology, Tuebingen (PACE is funded by the Baden-Wuerttemberg Ministry of Science, Research and the Arts). The funders had and will not have a role in study design, data collection and analysis, decision to publish, or preparation of the manuscript.

**Competing interests:** The authors have declared that no competing interests exist.

development. From childhood to adulthood, frontal activation in the inferior frontal gyrus (IFG) and middle frontal gyrus (MFG) decreases, while parietal activation in the intraparietal sulcus (IPS), supramarginal gyrus (SMG), and angular gyrus (AG) increases [6–10]. The brain areas in this fronto-parietal network fulfill different functions during mental arithmetic [4, 5]: Activation of the IFG and MFG is mainly associated with domain-general processes, such as working memory required for arithmetic; however, the exact location in the prefrontal cortex depends on the specific task demands [11]. Activation of the IPS, located at the border between the superior parietal lobule (SPL) and the inferior parietal lobule (IPL), reflects domain-specific number magnitude processing, and thus, an age-related activation increase indicates functional specialization [7]. Less deactivation of the SMG and AG (constituting the IPL) is specific for arithmetic fact retrieval, i.e., the automatic mapping of single-digit arithmetic problems to their solution [12]. Altogether, there is evidence for a developmental shift from domain-general frontal activation towards more specific parietal activation within the fronto-parietal network of arithmetic [6, 13], when comparing children and adults.

Moreover, the question arises how these developmental changes progress during aging. With increasing age, there is a decline in general cognitive capacities [14, 15]. Cognitive processing in elderly is especially limited by decreased working memory capacity and processing speed. Although some numerical abilities are affected by age-related changes, arithmetic skills are known to be mainly preserved in the elderly [16, 17, for a review see 18]. For instance, arithmetic fact knowledge and automated counting procedures in the single-digit range were shown to be equal or even superior to younger adults [18, 19]. Since neuroimaging research on arithmetic in the elderly has never been conducted so far, this project on the behavioral and neural correlates of arithmetic during development across the whole lifespan will open a new research field in numerical cognition. The investigation of the mechanisms underlying arithmetic in the elderly brain will provide insights into whether the developmental fronto-parietal shift continues during aging, and whether arithmetic is preserved during aging or shows deficits due to the general cognitive decline, thus requiring compensation.

When studying the underlying domain-general and domain-specific processes of arithmetic across the lifespan, the complexity of the arithmetic task plays an important role. Unfortunately, the majority of neuroimaging studies on arithmetic mainly focus on single-digit arithmetic, although after their first year of school, children mainly calculate with multi-digit numbers, and adults need multi-digit arithmetic in their daily lives well into old age, e.g., to handle finances, to compare prices while shopping, to manage their time, and to calculate weights while cooking. Knowledge of the place-value system is crucial for multi-digit number processing and consists of three levels: place identification, place-value activation, and place-value computation [20]. Multi-digit arithmetic is especially difficult when it requires operations across place-values, i.e., place-value computation [20]. This is the case for the carry operation in addition (when the sum of the units exceeds 9 so that the decade digit of the unit sum needs to be carried to the decade sum; e.g., 58 + 36 vs. 51 + 43) and the borrow operation in subtraction (when the minuend unit is smaller than the subtrahend unit so that a decade digit of the minuend needs to be borrowed for the difference of the units; e.g., 94–36 vs. 94–43).

The carry and borrow operations increase the difficulty of addition and subtraction problems as reflected by behavioral [21, 22] and neural effects [23–26]. The behavioral carry and borrow effects were already demonstrated in children [27–29], in adolescents [8, 30], in young adults [21, 22, 31], and in the elderly [32, 33]. The difficulty, as indicated by increases in reaction time and error rate, is attributed to higher working memory load, reflecting domain-general processes when comparing problems with carry and borrow operations to problems without them, i.e., the categorical carry and borrow effects [22, 34, 35]. While the categorical carry and borrow operations require place-value computation–the highest level of place-value

processing–the continuous carry and borrow operations instead require place-value activation [20]. Continuous processing characteristics of the carry operation are indicated by the unit sum (e.g., 58 + 36 has a unit sum of 8 + 6 = 14), which theoretically ranges from 0 to 18 (carry operation necessary when unit sum $\geq$ 10); continuous processing characteristics of the borrow operation are indicated by the unit difference (e.g., 94–36 has a unit difference of 4–6 = –2), which theoretically ranges from –9 to +9 (borrow operation necessary when unit difference < 0). The carry operation is characterized by both categorical and continuous processing characteristics [31, 36]. The nature of the borrow operation, however, remains unclear, since continuous processing characteristics might not necessarily increase the difficulty of subtraction in a similar way as it is the case for addition [37].

During development, children get better in arithmetic and place-value processing [38, for a review see 39]. Children learn the carry and borrow operations for single-digit and two-digit arithmetic in the first two years of elementary school and afterwards are able to use these skills [27]. Place identification, the first level of place-value processing, serves as a precursor for arithmetic performance and place-value computation later in elementary school [29]. The carry and borrow effects decrease during adolescence from grade 5 to 7 and to young adulthood [30], but not before [8, 27, 40]. Besides general increases in reaction times and error rates during aging, the carry and borrow operations are not impaired and might be even superior in older as compared to younger adults, as reflected by similar or smaller carry and borrow effects [32, 33, 41]. This reflects a general decrease in the carry and borrow effects during lifespan development with increasing proficiency in place-value processing. Furthermore, the underlying processing characteristics for these effects might change, since the carry and borrow effects seem to be categorical effects in elementary school children [27], while the carry effect in young adults relies on continuous processing characteristics as well, as assessed by the unit sum [26, 31, 36]. This suggests that primarily domain-general processes are driving the carry and borrow effects in children, while in young adults, domain-specific processes are additionally driving the carry effect. Considering the general cognitive decline during aging, the carry and borrow effects might be mainly driven by domain-specific processes in the elderly, but empirical evidence is still missing. The next step is now to replicate the developmental changes of arithmetic in general and place-value computation in particular (i.e., carry and borrow effects), to identify the underlying processing characteristics (i.e., categorical and continuous aspects) across the lifespan, and to complement the behavioral findings by neural data.

The neural representation of the carry and borrow effects is located in the fronto-parietal network of arithmetic processing [26]. Mainly, carry and borrow effects are associated with higher prefrontal activation in the left IFG and bilateral MFG [23–26, 42, 43], mostly reflecting domain-general demands like working memory for task difficulty due to the categorical effects [44, 45]. Additionally, parietal activation, particularly in the left IPS, was observed with increasing unit sum or when carry and borrow effects were confounded with problem size, mostly reflecting domain-specific magnitude processing associated with the continuous effects [23, 24, 26]. Furthermore, the only study on the neural correlates of the carry and borrow effects that was not conducted in young adults found the left AG to be reversely related to the carry effect in adolescents, reflecting the role of arithmetic fact retrieval for decomposed addition [8]. Taken together, the carry and borrow effects are associated with increases in frontal and parietal activation. Due to the developmental fronto-parietal shift in brain activation for arithmetic in general, the neural activation might also change for processing arithmetic complexity. For instance, frontal activation associated with the carry and borrow operations might decrease during lifespan development due to automatization, since the use of domain-general processes becomes more efficient from childhood to adulthood, and the general cognitive decline restricts further efficient use of domain-general processes during aging. On the other

hand, a study on interindividual differences in math ability showed that smaller behavioral carry and borrow effects might be associated with larger neural effects (in high- as compared to low-skilled individuals), since high-skilled individuals efficiently used the frontal resources for the carry and borrow operations whereas low-skilled individuals needed these resources even for problems without carry or borrow operation [25]. Thus, a decrease of the behavioral carry and borrow effects during lifespan development–with individuals getting better in the carry and borrow operations–might be associated with an increase in the neural carry and borrow effects in frontal brain regions. The current study aims to investigate the behavioral and neural correlates of the carry and borrow effects in children, young adults, and the elderly, and to explore developmental changes.

Arithmetic processing in general and the carry and borrow effects in particular are represented in a fronto-parietal network of arithmetic processing. Here, we address the question of how arithmetic and the underlying brain activation in the arithmetic network change during development across the lifespan. The study will target crucial stages of development: elementary school children in grades 3 and 4, because they just acquired the skills for two-digit arithmetic and thus serve as a starting point of lifespan development, young adults, because they have already established their arithmetic skills and thus serve as a reference for development, and the elderly, because they are experienced in using their arithmetic skills but might show a general cognitive decline and thus serve as a final point of lifespan development. A neural activation shift is hypothesized from frontal activation, mostly representing domain-general processes, to parietal activation, mostly representing domain-specific numerical processes, during development. Using three different approaches [29], we address the following hypotheses on behavioral and neural levels:

- H1: In a task-based approach, arithmetic performance is expected to be better in young adults than in children and in the elderly. According to the developmental fronto-parietal shift for arithmetic processing, children should show increased frontal activation (left IFG and bilateral MFG) and less parietal activation (left IPS) in comparison to young adults, replicating previous research. Moreover, the neural correlates of arithmetic will be explored in the elderly: if the developmental fronto-parietal shift generalizes to the whole lifespan, the elderly might show less frontal activation (left IFG and bilateral MFG) and increased parietal activation (left IPS) in comparison to young adults; however, if arithmetic is not affected by aging or is affected and needs compensation, the elderly might show similar or increased frontal activation and similar or less parietal activation in comparison to young adults.

- H2: In an effect-based approach, the carry and borrow effects are expected to decrease arithmetic performance on a behavioral level (i.e., reaction times and error rates) and to be associated with larger frontal activation (left IFG and bilateral MFG) on a neural level in all age groups. Regarding the lifespan development, the behavioral carry and borrow effects are expected to be larger in children than in young adults and larger or similar in young adults as compared to the elderly, replicating previous research. The neural development of the effects will be explored here for the first time: the neural carry and borrow effects might either decrease during the lifespan, reflecting the fronto-parietal activation shift particularly for complex arithmetic, or increase during the lifespan, reflecting the more efficient use of frontal resources with increasing performance.

- H3: In an effect-based approach on the underlying processing characteristic, the carry and borrow effects are expected to be rather categorical in children, both categorical and continuous in young adults, and rather continuous in the elderly. On a neural level, the continuous

carry and borrow effects (unit sum and unit difference) should rely on parietal activation (left IPS) in all age groups.

The carry and borrow effects will be investigated concerning arithmetic complexity in two-digit addition and subtraction, respectively. The processes underlying these arithmetic operations are similar, while subtraction is more difficult than addition [25]. However, there is not yet evidence for operation-specific differences in relation to lifespan development. To assess brain activation during two-digit arithmetic, the optical neuroimaging method fNIRS will be used. Compared to fMRI, fNIRS is less restrictive, allows an upright body position, and is relatively insensitive to motion artifacts–but at the cost of a lower spatial and depths resolution [46]. The current study makes use of the advantages of fNIRS to study arithmetic in an ecological valid task paradigm (verbal production) in critical populations such as children and the elderly.

## Methods

### Participants

Three age groups will be considered: children (3rd and 4th grade), young adults (18–34 years), and the elderly (above 60 years). Each age group will be characterized by age (*M*, *SD*, *Range*), gender, and education. All subjects will be right-handed, native German speakers (or at least school education in German), with no history of neurological or mental disorders, and without a disease that influences brain metabolism. Informed consent will be obtained from all adult participants, from the parents of the participating children, and from the participating children in a simplified way. For participation, all subjects will receive monetary compensation and the children additionally a little present. The study was approved by the Ethics Committee for Psychological Research of the University of Tuebingen.

To emphasize the focus on healthy aging in the current study, elderly subjects will additionally be assessed by the Montreal Cognitive Assessment [MoCA; 47], which is a brief cognitive screening tool for mild cognitive impairment. This instrument measures cognitive abilities such as short-term memory, visuo-spatial and executive functions, attention, language, and orientation to time and space. A cut-off score of $\geq$ 26 (*Theoretical Range* = 0–30 with correction in case of education years $\leq$ 12) will be used to exclude cognitive impairment in elderly. Additionally, processing speed, working memory, and verbal and non-verbal intelligence will serve as control measures to compare general cognitive abilities between the age groups.

### Arithmetic task

The arithmetic task will consist of two-digit addition and subtraction problems with two operands resulting in a two-digit solution [https://osf.io/6emdy/]. The carry and borrow operations will be manipulated categorically, i.e., addition problems with and without carrying (e.g., 51 + 43 vs. 58 + 36) as well as subtraction problems with and without borrowing (e.g., 94–43 vs. 94–36), and continuously, i.e., unit sum in addition (e.g., 14 in 58 + 36) and unit difference in subtraction (e.g.,– 2 in 94–36).

The stimulus set consists of 128 arithmetic problems with 32 trials per condition, i.e., addition with/without carrying and subtraction with/without borrowing. The stimulus generation will consider unit sum in addition and unit difference in subtraction to be relatively equally distributed. This means that addition problems ($a + b = r$) cannot be directly transformed by inversion into subtraction problems ($r–b = a$). Nevertheless, the numerical properties will be matched across conditions: *a* and *b* will be closely matched in their numerical magnitude and

parity; the position of the larger operand will be counterbalanced; pure decades as well as ties within and between *a*, *b*, and *r* will be excluded [25, 42, for decades see 48, for ties see 49].

The task will be computerized in the program OpenSesame [50]. Each arithmetic problem will be presented centered in white against a black background. In an oral production paradigm, the subjects will be asked to mentally solve the problem as quickly and accurately as possible and to respond while pressing the space bar. Button press and button release will be recorded and the experimenter will blindly note the given responses. Each stimulus will be presented with a time limit of 30 s and disappear upon button press to emphasize mental arithmetic before responding. In the inter-trial interval, a black screen will be shown with a duration of 4–7 s (jittered in steps of 0.5 s, mean of 5.5 s), including a white fixation point in the last 0.5 s. The 128 trials will be presented in 4 runs of 32 trials each (8 trials per condition) and the trial order will be pseudorandomized for every subject with no more than two trials of the same condition presented consecutively. As dependent variables, error rates (ER) are defined as the number of incorrectly solved and time-out trials divided by the total number of completed trials, and reaction times (RT) as the duration between stimulus onset and button press.

### Cognitive tests

Processing speed will be assessed by the subtest symbol search of the German version of the Wechsler Adult Intelligence Scale IV [WAIS-IV; 51]. In this paper-pencil test, subjects will be asked to decide whether or not two target symbols are present among a group of five symbols. The overall time limit is 120 s for a maximum of 60 items. The raw score (*Theoretical Range* = 0–60) is defined as the number of correctly solved items minus the number of incorrectly solved items (unsolved items are not considered). The retest reliability of the German version is good ($r_{tt}$ = .81).

Working memory will be assessed by a verbal 2-back paradigm with letters [https://osf.io/6emdy/; 52]. The task is computerized in the program OpenSesame [50]. In this task, subjects will be asked to determine for every letter (consonants in lower and upper case) whether it matches the letter presented two positions before or not by a button press. The 92 trials consist of 30 match trials (match to the letter presented two positions before), 48 mismatch trials (no match to a letter presented one to five positions before), 6 1-back lures (match to the letter presented one position before), 6 3-back lures (match to the letter presented three positions before), and 2 start letters (presented first). Each stimulus will be presented until the button press with a time limit of 2.5 s followed by an SOA of 3 s. Accuracy (ACC) will be defined as the number of correct trials divided by the total number of trials, and RT as the duration between stimulus onset and button press. Note that the comparison between the groups will focus on ACC.

Intelligence will be assessed by the German version of the Reynolds Intellectual Screening Test [RIST; 53], consisting of subtests for verbal and nonverbal intelligence. Depending on age, each subtest is started at a certain item (with the option of going back to preceding items until two consecutive items are correctly solved in the first attempt) and stopped when three consecutive items are not correctly solved. In the subtest "guess what", indicating verbal intelligence, subjects are orally asked to find out the concept that matches the given two to four clues. With a maximum of 62 orally presented items, the raw score (*Theoretical Range* = 0–62) is defined as the sum of all correctly solved items (including the unsolved items before the age-dependent start item). In the subtest "odd item out", indicating nonverbal intelligence, subjects are asked to choose the picture that does not belong to the set of five to seven pictures. The time limit is 30 s for the first attempt and 20 s for the second attempt (if incorrectly or unsolved in the first attempt), with a maximum of 51 visually presented items. The raw score

(*Theoretical Range* = 0–102) is calculated as the double sum of all correctly solved items in the first attempt (including the unsolved items before the age-dependent start item) and the single sum of all correctly solved items in the second attempt. Raw scores for both subtests will be further transformed into *T* scores (*M* = 50, *SD* = 10) dependent on German age norms and converted into IQ scores (*M* = 100, *SD* = 15). The reliability of the German version is good (Cronbach's $\alpha$ of .92 for verbal intelligence and .90 for nonverbal intelligence).

## Procedure

During the fNIRS measurements, the arithmetic task will be conducted in a light-attenuated room. Afterwards, processing speed, working memory, and intelligence will be assessed. Each test is preceded by instructions and practice items. The practice phase of the arithmetic task will consist of 12 trials to familiarize the subjects with the response format (which can be repeated). In the end, a screening of cognitive abilities will be conducted for older adults only.

## fNIRS data acquisition

The fNIRS data will be acquired using the continuous wave ETG-4000 Optical Topography System (Hitachi Medical Corporation, Tokyo, Japan). This fNIRS device uses wavelengths of 695 ± 20 nm and 830 ± 20 nm as light sources and a sampling rate of 10 Hz. The optodes (10 sources and 8 detectors) will be embedded in a cap (Brain Products GmbH, Herrsching, Germany) with an inter-optode distance of 30 mm. The probesets will consist of 4 parietal channels (IPS, SMG, AG) and 5 frontal channels (MFG, IFG) per hemisphere (see Fig 1), which is a

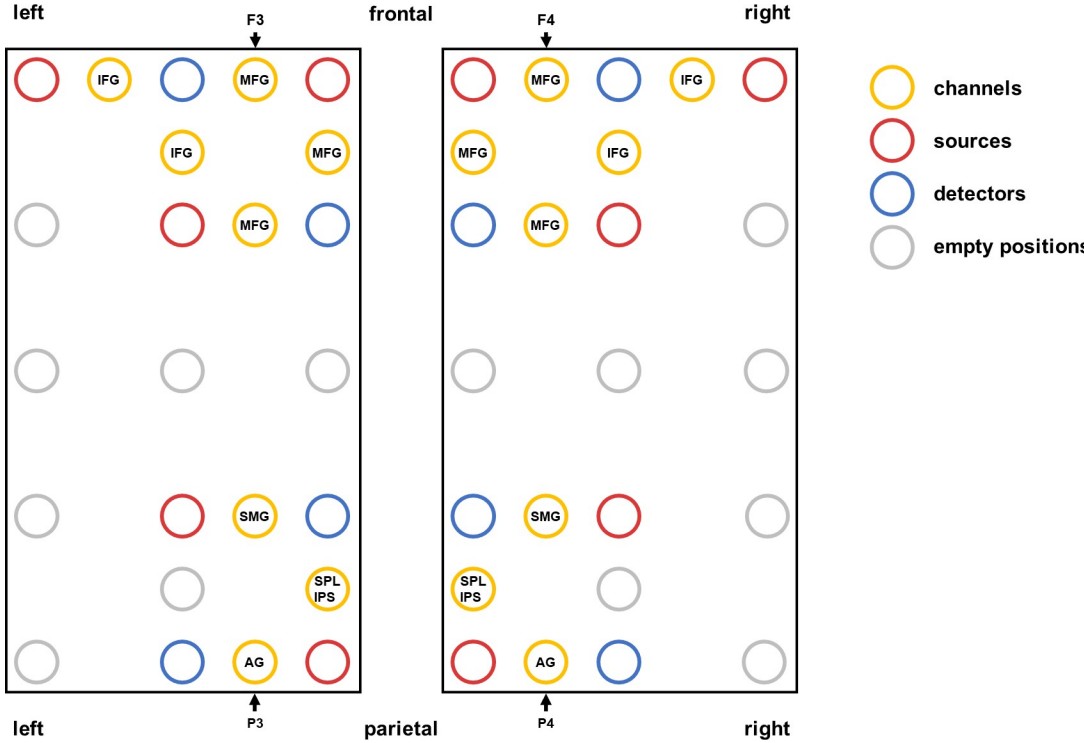

**Fig 1. fNIRS probesets covering frontal and parietal areas of the left and right hemisphere.** The probesets were fixed at P3/P4 and oriented towards F3/F4; the positions of channels and optodes (sources and detectors) including empty positions are marked [cf. 25]. *Abbreviations of the channel labels*: IFG–inferior frontal gyrus, MFG–middle frontal gyrus, SPL/IPS–superior parietal lobule/intraparietal sulcus, SMG–supramarginal gyrus, AG–angular gyrus.

subset of channels that were previously used as a probeset [for more details on the location about the probeset see 25]. The correspondence of fNIRS channels to the underlying cortical areas was estimated based on a virtual registration method [54–56] and labeled according to the automated anatomic labeling (AAL) atlas [57].

## Data analysis

**Data exclusion for subjects.** Subjects will be excluded from all analysis when the inclusion criteria are not met (regarding age, handedness, language, health, and, for elderly subjects, cognition), when more than 50% of the behavioral data of the arithmetic task is missing (due to drop out or experimental or technical problems), or when the error rate in the arithmetic task is larger than 50% (because of the exclusion of incorrectly solved trials). Moreover, an outlier analysis will be conducted specific to the respective age group to exclude subjects deviating more than 3 median absolute deviations from the group's median in RT of the arithmetic task from all analysis [58]. Subjects will be excluded only from neural data analysis in case of more than 50% missing neural data of the arithmetic task (due to drop out, experimental or technical problems, trial exclusion and artifact rejection), or in case of more than 3 noisy channels (restricting channel interpolation to a maximum of 3 channels). Furthermore, a case-wise exclusion of subjects from the respective analysis of the demographic variables (age, gender, education) or control measures (processing speed, working memory, and verbal and non-verbal intelligence) applies in case of missing or incomplete data, or an ACC below 33% in the working memory task.

**Data exclusion for trials.** Trials will be excluded from the RT analysis (arithmetic and working memory task) as well as from the fNIRS analysis (arithmetic task) when the trial was not correctly solved, when the RT was below 200 ms (anticipations), when the RT deviates more than 3 median absolute deviations from the subject's median in the respective task, and when the duration between button press and button release deviates more than 3 median absolute deviations from the subject's median in the arithmetic task [58].

**fNIRS data preprocessing.** The relative concentration changes of oxygenated ($O_2$Hb) and deoxygenated hemoglobin (HHb) will be calculated for every fNIRS channel. The fNIRS signal will be preprocessed by using the temporal derivative distribution repair [TDDR; 59] to correct for high-amplitude motion artifacts and by applying a bandpass filter of 0.005–0.2 Hz. To reduce low-amplitude motion artifacts, correlation-based signal improvement [CBSI; 60] will be used, which is based on the negative correlation between $O_2$Hb and HHb and is considered one of the best artifact correction methods [61]. Next, remaining noisy channels will be interpolated by surrounding channels, and incorrectly solved trials as well as trials containing uncorrectable artifacts will be excluded.

To analyze the fNIRS data within a model-based approach, the peak latency of the hemodynamic response function will be determined by the overall maximum across channels, subjects, and conditions [in the interval between 4 and 10 s rounded to half a second; 25]. In the model-based approach, a general linear model will be computed for each channel, subject, and condition according to the hemodynamic response function. For every region of interest (10 ROIs: IFG, MFG, IPS, AG, SMG on the left and right hemisphere), the channel with the highest resulting beta value based on the grand average across conditions and subjects will be used for the statistical analysis of the neural data.

**Statistical data analysis.** This study applies Bayesian hypothesis testing and thus Bayes factors (*BF*) are calculated that determine how much more likely the observed data will be under the alternative hypothesis ($H_1$) as compared to the null hypothesis ($H_0$) for $BF_{10}$ (evidence for a difference when $BF_{10} > 1$) and vice versa for $BF_{01}$ (evidence for no difference when

$BF_{01} > 1$), whereby $BF_{01} = 1/BF_{10}$ [62]. $BF$s can be interpreted to provide anecdotal evidence for 1–3, moderate evidence for 3–10, strong evidence for 10–30, and very strong evidence for 30–100, and extreme evidence above 100 in favor of one hypothesis [63, 64].

The statistical analyses in terms of Bayesian $t$-tests, Bayesian ANOVAs and Bayesian linear regressions will be performed with JASP (Jeffreys's Amazing Statistics Program, JASP Team, 2016). The analysis prior will be set to a Cauchy prior scale of 0.707 in Bayesian $t$-tests, reflecting that $H_0$ and $H_1$ are equally likely to occur, to the default Cauchy prior of $r = 0.5$ for the fixed effects in Bayesian ANOVAs, to the default Jeffreys–Zellner–Siow prior of $r = 0.354$ for regression coefficients, and to a uniform model prior in Bayesian linear regressions. Bayesian ANOVAs and linear regressions will set out to compare each model to the null model and Bayesian model averaging will compare the models with the respective effect to equivalent models without the effect (analysis suggested by Sebastiaan Mathôt).

Prior to the analyses of the arithmetic task, cognitive abilities will be analyzed: children and the elderly will be compared to young adults regarding processing speed (raw scores), working memory (ACC), and verbal and non-verbal intelligence (IQ scores) by two-sided Bayesian independent samples $t$-tests. Next, the arithmetic task will be analyzed in 3 age (children, adults, elderly) × 2 operation (addition, subtraction) × 2 complexity (with, without carry/borrow) Bayesian repeated measures ANOVAs [analysis over subjects, averaged over trials]. For effects with $BF_{10}$ or $BF_{01} \geq 6$, post-hoc two-sided tests will be conducted for the contrast children vs. young adults and the contrast young adults vs. the elderly by means of Bayesian independent samples $t$-tests, or for contrasting different conditions by means of Bayesian paired $t$-tests. Finally, multi-model Bayesian regressions will be conducted with the categorical predictor carry/borrow (with, without) and the continuous predictor unit sum/difference separately for addition and subtraction and for every age group [analysis over trials, averaged over subjects]. All confirmatory analyses will be conducted on the dependent variables RT, indicating behavioral performance, and beta values, indicating neural activation for each ROI separately (left/right IFG, MFG, IPS). Planned exploratory analyses will be conducted on ER and beta values for neural activation in the other ROIs (left/right AG, SMG).

## Bayes factor design analysis

For sample size estimation, the sequential Bayes factor design with maximal $n$ will be used [65, 66]. In this design, data collection (1) will start with a minimum sample size of $n_{min} = 20$ per group, (2) will continue until a $BF_{10}$ or $BF_{01} \geq 6$ is obtained for all effects of interest, or (3) will be stopped when a maximum sample size of $n_{max} = 60$ per group has been reached. The properties of the planned research design were estimated with Monte Carlo simulations according to Schönbrodt and Wagenmakers [65]: The minimum sample size was set for reducing false positive rates, and the maximum sample size was set to ensure feasibility, while 80% of studies with an infinite sequential sampling stop earlier than $n_{max}$. If sampling is terminated because of reaching $n_{max}$, only with a probability of 5% will the study obtain misleading evidence. This design detects an expected medium effect size of $\delta = 0.5$ with a probability of 61% before the $n_{max}$ is reached. The chosen medium effect size accounts for the bias of small samples in the reported large effect sizes ($\eta_p^2 \approx .5$) for differences between children, young adults, and the elderly in the neural distance effect [67, for large effect sizes ($d \approx 1.2$–$1.6$) see also for children vs. adults: 68, for younger adults vs. the elderly: 69].

The effects of interest according to the hypotheses include the main effect of age (ANOVAs) for RT and activation in left IFG, bilateral MFG, and left IPS according to H1; the main effect of complexity and the interaction effect of complexity and age (ANOVAs) for RT and activation in left IFG and bilateral MFG according to H2; the categorical carry/borrow effect and the

continuous effect of unit sum/difference in each age group (regressions) in RT and neural activation in left IPS according to H3.

## Proposed timeline

After in-principle-acceptance of the registered report in stage I, data collection can start whenever the current global pandemic situation permits testing with children and elderly subjects. Data collection is estimated to last 1 year (planned for September 2021 –August 2022), followed by approximately 3 months for data analysis and preparation of the registered report for stage II (planned for September–November 2022).

## Acknowledgments

I would like to thank Isabel Kriechel, Milena Glueck, Merle Bode, and Stephanie Fengler for preparations of the study, and Morgan Hess for language proof-reading of the manuscript. I acknowledge support by Open Access Publishing Fund of University of Tuebingen.

## Author Contributions

**Conceptualization:** Christina Artemenko.

**Funding acquisition:** Christina Artemenko.

**Investigation:** Christina Artemenko.

**Methodology:** Christina Artemenko.

**Project administration:** Christina Artemenko.

**Supervision:** Christina Artemenko.

**Writing – original draft:** Christina Artemenko.

**Writing – review & editing:** Christina Artemenko.

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
