## [Decision Letter · Decision Letter 0]

13 May 2021

PONE-D-21-08537

Developmental fronto-parietal shift of brain activation during mental arithmetic across the lifespan

PLOS ONE

Dear Dr. Artemenko,

Thank you for submitting your registered report to PLOS ONE. I have sent it to two expert reviewers and have received their comments. As you will see at the bottom of this email, both reviewers found that the report was well motivated, well written and the proposed methodology sound and able to answer your hypotheses. I do agree with the reviewers on all of these points. Both reviewers, however, have a number of suggestions. I will not reiterate these here as the reviewers did a careful evaluation of your manuscript and the comments are quite clear. But I encourage you to take these into account in a revision of the report.

We look forward to receiving your revised manuscript.

Kind regards,

Jérôme Prado

Academic Editor

PLOS ONE

Journal Requirements:

As this manuscript is a Registered Report Protocol, please ensure that you indicate this in your title.

In your Methods section, please state how you will obtain consent. Specifically, please state whether you will obtain consent from parents or guardians of the minors included in the study or whether the research ethics committee or IRB specifically will waive the need for their consent.

Reviewers' comments:

Reviewer's Responses to Questions

6. Review Comments to the Author

You may also provide optional suggestions and comments to authors that they might find helpful in planning their study.

Reviewer #1: This study aims to investigate if developmental changes observed between childhood and young adulthood in the relative engagement of frontal and parietal regions during mental arithmetic would show continuity or change in older adulthood. Using a cross-sectional approach, the study will characterize the behavioral and neural effects during multi-digit mental arithmetic across different age groups. The lifespan approach is valuable, especially in light of the decline in general cognitive capacities in old age that may affect how mental arithmetic is accomplished in older adults to maintain the same level of performance. The focus on multi-digit mental arithmetic is practically motivated and a theoretically appropriate one given that it would tap heavily into the engagement of frontal regions implicated in working memory and executive functions. These cognitive constructs underlie the concept of the fronto-parietal shift. In general, the study is well motivated, and the protocol is sound and suitable for addressing the hypotheses. I am confident that the author will be able to execute the project successfully. The current version of the registered report is already very well written, but it would be great to have the author clarify and justify the hypotheses and predictions a little more. Below are my specific comments about these issues, as well other theoretical and methodological ones:

1. This first point is a general observation of how the literature has characterized the so-called fronto-parietal shift. Based on my own reading of the literature, the frontoparietal network is engaged during mental arithmetic even in adults (e.g., in an empircal study by Grotheer et al., 2019; or in a meta-analysis by Hawes et al., 2019). However, from the second paragraph on page 3, I am afraid that readers may get the impression that the putative shift is qualitative rather than quantitative. By qualitative, I mean there tends to be a simplified and inaccurate understanding that adults rely mainly on the parietal regions and less (or not at all) on frontal regions; and the reverse applies to children. By quantitative, I mean that whenever researchers present activation maps for children and adults separately, the fronto-parietal network is recruited regardless of age, but the “shift” emerges only when the maps are contrasted between age groups. I am not sure if the author shares the same sentiment, but it would be helpful if the author can be clearer about describing the so-called shift.

2. The author introduces task complexity as an additional layer of complication to the developmental fronto-parietal shift. It is definitely an interesting dimension, but it can be challenging to interpret the findings. As both the shift and task complexity implicate the frontal regions, the report at this stage will benefit from a consideration whether the effects are additive, or whether they interact and how they may interact. This will lead to more specific hypotheses, and also a better delineation between confirmatory and exploratory analyses. In the current version, the author does not indicate if any anticipated or post-hoc exploratory analyses will be performed.

3. Please clarify and elaborate on what is meant by “continuous processing characteristics of the carry effect” on page 4. Although the author alluded to a distinction between “categorical” (i.e., with and without carry or borrow operations) and “continuous”, it is not immediately clear whether “continuous” (as assessed based on unit sum) refers to within each condition, just within the condition with carry/borrow operations, or across conditions (i.e., collapsed across all trials). Readers who are not familiar with previous work (and would not have read the methods section) may not be familiar with the distinction and why they matter. It would therefore be necessary to elaborate on these types of processing characteristics by providing more concrete examples.

4. In several paragraphs on page 4 and beyond, I believe that “carry/borrow effects” might have been mistakenly used in place of “carry/borrow operations”. For instance, on page 4, “rely on their initial place-value understanding for addition performance and the carry effect later in elementary school”. An “effect” as described earlier in the introduction refers to the difference in performance between two conditions (with and without carry/borrow operations). It is also not immediate clear to me the distinction between “place-value understanding” and carry operation in this context. The clarity of the paper can be enhanced if the author could check the specificity and intended contextualized meanings of these terms. The author may also choose to provide their own nuanced definitions.

5. The author noted that frontal involvement found in the carry and borrow effects reflects “domain-general demands like working memory for task difficulty due to the categorical effects” (p. 5). However, frontal involvement may not always reflect domain-general demands (e.g., see Fedorenko & Blank, 2020 for prefrontal involvement specific to language or multiple demands; see Sokolowski et al., 2017 for frontal involvement across many basic numerical tasks). The association between frontal involvement and domain-general demands is a common reverse inference, but does not appear to have strong empirical evidence for. Would the author consider including an independent working memory task (or a relevant task) to directly test for the hypothesis of working memory/domain-general demands so as to strengthen the inferences that could be made from this study? This may also help clarify whether parietal activation truly reflects “domain-specific magnitude processing associated with the continuous effects” (p. 5), or can be partly accounted for by working memory (because unit sums can also involve procedural computation rather than fact retrieval).

6. For H1, please clarify and justify the hypothesis that “children should mainly rely on frontal regions (left IFG and bilateral MFG), whereas elderly should mainly rely on parietal regions (left IPS) in comparison to young adults” (p. 5). In particular, what does “mainly rely on” means. It can be interpreted as “only involves”. Alternatively, the author may mean “should recruit to a greater extent frontal/parietal regions”. Also, why would that be the case for the elderly? Is it possible that there is no quantitative difference between the elderly and young adults? Could the elderly instead recruit frontal regions to a greater extent than young adults (i.e., children-like), independent of what is observed for the parietal activation?

7. For H2, could the behavioral carry and borrow effects be equivalent between young adults and elderly as well? This alternative hypothesis seems plausible given that previous research found that “carry and borrow operations are not impaired and might be even superior in older as compared to younger adults, as reflected by similar or smaller carry and borrow effects” (p. 4).

8. For H2, the author also hypothesized that “neural development of effects… might decrease during the lifespan due to the fronto-parietal activation shift for arithmetic in general” (p. 5). It is not immediately clear what this means, and it is also made complicated by the conflation of task-based and effect-based neural effects. Are these neural effects additive, or do they interact? It would be good to unpack this more carefully so that the planned analyses can be better evaluated. To address this, the author should provide clearer hypotheses about how the extent of the frontal and parietal activations may manifest in the age-by-condition (with and without carry/borrow operations) interaction in relation to the fronto-parietal shift framework. Finally, related to the other issues brought up, it is not clear what the author means by “when considering interindividual performance differences where smaller behavioral carry and borrow effects were associated with larger neural effects for frontal activation due to math ability, the neural carry and borrow effects might also increase during lifespan” (p. 6). It is not apparent why individual differences are considered here when the primary questions are about group averages. Also, why would smaller behavioral carry and borrow effects be associated with larger frontal activation? It would help with understanding the logic if the author could unpack these parts a little more.

9. For H3, it is not immediately clear based on the discussion in the introduction why the carry and borrow effects will no longer be categorical in elderly. This is especially so based the point raised in Comment #7. Is there a prediction for the development of parietal activation for the continuous carry and borrow effects?

10. For clarity, please justify why 3rd and 4th grades will be used. Is it because multi-digit arithmetic instruction only starts in 2nd grade?

11. The use of sequential Bayes factor design used to estimate sample size is commendable and well-described. Please provide citation(s) for “80% of studies with an infinite sequential sampling stop earlier than nmax. If sampling is terminated because of reaching nmax, only with a probability of 5% the study will obtain misleading evidence”, or whether a simulation was conducted.

12. The arithmetic task involves both addition and subtraction that will be analyzed as separate levels of the operation factor. However, H1 – H3 mainly focus on age, complexity, and age-by-complexity interaction. Hence, at present, the hypotheses (nothing about operation per se) and analyses (operation as a factor of interest) are not congruent. Please reconcile these differences as appropriately guided by theory (e.g., provide hypotheses about three-way interactions, focus on one operation, or collapse across operations for the primary planned analyses, but analyze them separately for planned exploratory analyses, etc.).

13. For measuring reaction time of mental arithmetic using button press, how will the author account for instances in which the button presses are accompanied by irrelevant verbalizations (e.g., “erm”) or followed by self-corrections? How reliable is this procedure compared to audio recording of verbal responses and manually extracting reaction times thereafter?

14. For outlier detection, the author may want to consider using the median absolute deviation instead of standard deviation based on the mean. See Leys, C., Ley, C., Klein, O., Bernard, P., & Licata, L. (2013). Detecting outliers: Do not use standard deviation around the mean, use absolute deviation around the median. Journal of Experimental Social Psychology, 49(4), 764–766. https://doi.org/10.1016/j.jesp.2013.03.013.

15. Why is SMG a recorded channel, but not a region of interest? If it is used for exploratory analyses, please describe what the analyses may be.

16. It appears that BF > 3 is used as a cutoff for inferences, but the Bayes factor design analysis used BF = 6 as a criterion. Please reconcile this difference. Please also provide the scale values for all default priors used, e.g., the width of the Cauchy prior for t-tests (0.707 is the default in JASP).

17. The author may want to consider using mixed models instead of repeated-measures ANOVA and simple regression to better handle missing data or even use trial level data instead of averaging across trials. Bayesian mixed models can be conducted in JASP. This is just a suggestion, and the author’s choice will not affect my recommendation in any way.

18. In the final paragraph of “Statistical data analysis”, ER was not mentioned as a dependent variable in all the effects of interest, contrary to the last sentence of the preceding paragraph. Please reconcile this difference.

19. Will other cognitive measures, such as processing speed, working memory, etc., be included as covariates in the primary analyses, especially if they differ between the age groups?

Minor Comments

1. In the abstract, “borrow” might be missing from “carry and operations”.

2. The full labels associated with some acronyms are not provided, e.g., SPL and IPL in the second paragraph of page 3.

3. The timeline may be a little optimistic given that a maximum of 60 subjects per group is expected to be collected. I suggest allocating more time for data collection.

References

Fedorenko, E., & Blank, I. A. (2020). Broca’s Area Is Not a Natural Kind. Trends in Cognitive Sciences, 1–15. https://doi.org/10.1016/j.tics.2020.01.001

Grotheer, M., Zhen, Z., Lerma-Usabiaga, G., & Grill-Spector, K. (2019). Separate lanes for adding and reading in the white matter highways of the human brain. Nature Communications, 10(1), 3675. https://doi.org/10.1038/s41467-019-11424-1

Hawes, Z., Sokolowski, H. M., Ononye, C. B., & Ansari, D. (2019). Neural underpinnings of numerical and spatial cognition: An fMRI meta-analysis of brain regions associated with symbolic number, arithmetic, and mental rotation. Neuroscience and Biobehavioral Reviews, 103(May), 316–336. https://doi.org/10.1016/j.neubiorev.2019.05.007

Sokolowski, H. M., Fias, W., Mousa, A., & Ansari, D. (2017). Common and distinct brain regions in both parietal and frontal cortex support symbolic and nonsymbolic number processing in humans: A functional neuroimaging meta-analysis. NeuroImage, 146(October 2016), 376–394. https://doi.org/10.1016/j.neuroimage.2016.10.028

Reviewer #2: The protocol looks rigorous and I think that it has potential to contribute to the literature. I don't see any major issues with this registered report. I only have several minors issues that might need to be clarified:

1. The introduction is very well written, but I found it difficult to understand exactly what was the difference between categorical and continuous borrows/carries. I am not sure whether this just refers to a methodological way of categorizing the stimuli (i.e. two binary groups vs. taking into account more stimulus properties) and/or correspond to measurable cognitive processes. For instance, the author refers to studies showing that the effect of borrow/carry is categorical in elementary school. Could we conclude that the borrow/carry effect is so strong within young children that it "hides" effects other stimulus properties? By extension, could we conclude that this carry/borrow effect would decrease with age, so that other effects (such as the unit sum) are "visible"? I would advise the author to clarify this distinction, since it is crucial for their later analyses. Optionally, the author might want to provide some examples or a categorical vs. continuous carry/borrow processes.

2. Relative to H1, "whereas the elderly should mainly rely on parietal regions (left IPS) in comparison to young adults." It seems to me that the author considers that the shift "continues" until old age. I am not sure that there are experimental evidence for this. It could alternatively that the shift is mature at adulthood and stay stable across life. That being said, it might be that elderly show less frontal activation (relative to young adults) due to general decline in frontal executive processes. Could the author specify/justify whether they expect old participants to rely less on frontal region due to the shift or to general cognitive decline in executive functions?

3. The author state that data collection "(2) will continue until a BF10 or BF01 of 6 is obtained". There are three main hypotheses, with several sub-hypotheses. Which hypothesis/es would be the criterion for this BF value?

4. Could you specify what is the maximum number of channels to be interpolated for each individual?

5. Very very minor: please provide the full name of SPL and IPL acronyms in the introduction

7. PLOS authors have the option to publish the peer review history of their article (what does this mean?). If published, this will include your full peer review and any attached files.

Reviewer #1: No

Reviewer #2: **Yes: **Mathieu Guillaume

---

## [Author Response · Author response to Decision Letter 0]

7 Jun 2021

The Response Letter is separately attached.

---

## [Decision Letter · Decision Letter 1]

3 Aug 2021

Developmental fronto-parietal shift of brain activation during mental arithmetic across the lifespan: A Registered Report Protocol

PONE-D-21-08537R1

Dear Dr. Artemenko,

I am pleased to inform you that your manuscript has been judged scientifically suitable for publication and will be formally accepted for publication once it meets all outstanding technical requirements.

Kind regards,

Jérôme Prado

Academic Editor

PLOS ONE

Additional Editor Comments (optional):

Reviewers' comments:

Reviewer's Responses to Questions

**Comments to the Author**

1. Does the manuscript provide a valid rationale for the proposed study, with clearly identified and justified research questions?

Reviewer #1: Yes

Reviewer #2: Yes

2. Is the protocol technically sound and planned in a manner that will lead to a meaningful outcome and allow testing the stated hypotheses?

Reviewer #1: Yes

Reviewer #2: Yes

3. Is the methodology feasible and described in sufficient detail to allow the work to be replicable?

Reviewer #1: Yes

Reviewer #2: Yes

4. Have the authors described where all data underlying the findings will be made available when the study is complete?

Reviewer #1: Yes

Reviewer #2: Yes

5. Is the manuscript presented in an intelligible fashion and written in standard English?

Reviewer #1: Yes

Reviewer #2: Yes

6. Review Comments to the Author

You may also provide optional suggestions and comments to authors that they might find helpful in planning their study.

Reviewer #1: The author has addressed all of my concerns satisfactorily. I look forward to reading about the findings.

Reviewer #2: I would like to thank the author for their thorough response letter, in which they clarify all my comments. I believe that the report in its current form is suitable for publication and I am recommending its acceptance.

7. PLOS authors have the option to publish the peer review history of their article (what does this mean?). If published, this will include your full peer review and any attached files.

Reviewer #1: No

Reviewer #2: No

---

## [Editor Report · Acceptance letter]

16 Aug 2021

PONE-D-21-08537R1 

Developmental fronto-parietal shift of brain activation during mental arithmetic across the lifespan: A Registered Report Protocol 

Dear Dr. Artemenko:

I'm pleased to inform you that your manuscript has been deemed suitable for publication in PLOS ONE. Congratulations! Your manuscript is now with our production department. 

Kind regards, 

on behalf of

Dr. Jérôme Prado 

Academic Editor

PLOS ONE